

# Vertical and temporal fluid mud dynamics during spring-neap tidal cycles

Aron Slabon[1], Lorenzo Rovelli[1], Dörthe Holthusen[1], Jill Lehn[1], Annika Fiskal[1], Ole Rössler[1], Christine Borgsmüller[1], Thomas O. Hoffmann[1]

5  [1]Federal Institute of Hydrology, 56068 Koblenz, Germany

*Correspondence to*: Aron Slabon (slabon@bafg.de)

**Abstract.** The hyper-turbid Ems estuary has undergone extensive channel deepening since the 1980s resulting in distinct tidal asymmetry and substantial sediment accumulation which led to the occurrence of a fluid mud layer. This might cover up to 60% of the water column, creating density-driven stratification that significantly affects hydrodynamics, ecology, and

10  navigability. During a six-week monitoring period we investigated fluid mud dynamics with the focus on occurrence, thickness, density, and driving processes to obtain insights about formation and break-up of the fluid mud layer. We identified two superimposed recurring cycles of fluid mud occurrence. On a shorter timescale, we observed distinct differences over the semidiurnal tide while on a longer timescale fluid mud occurrence shows a distinct spring-neap tide relation. We conducted dedicated measurement campaigns around spring and neap tide to further investigate spring-neap variability of fluid mud

15  occurrence. One main finding of those campaigns is that during neap tide, the majority of the water column is covered by a dense fluid mud layer, thus reducing the hydrodynamic cross-section. This leads to increased current velocity and entrainment at the lutocline, and reduced stability of the fluid mud layer. In contrast during spring tide, the hydrodynamic cross-section remains wider, leading to less friction at the lutocline and thus enhances stability of the fluid mud layer and thus longer persistence of fluid mud.

**Graphical abstract / Key figure.**

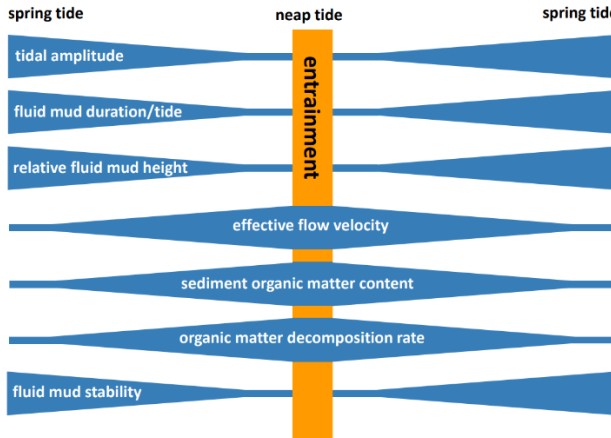





**Short summary.** In the Lower Ems estuary, ships and ecosystems face challenges from dense layers of sediment known as
fluid mud. Using long-term, high-resolution observations, we found that spring tides promote stable fluid mud layers, while
during neap tides they are less stable and are more likely to disperse. These dynamics shape sediment movement, affect
navigation, and influence the ecosystem, highlighting why understanding fluid mud is essential for managing hyper-turbid
estuaries.

# 1 Introduction

Hyper turbidity, i.e., the occurrence of extremely high sediment concentrations (up to several g L$^{-1}$), affects many coastal and
estuarine areas worldwide (Wang et al., 2020; Papenmeier et al., 2012; Schulz et al., 2022). This phenomenon is caused by
natural processes such as storms and river discharge, and by human activities like dredging and construction (Oberrecht and
Wurpts 2014). High sediment concentrations hinder ship navigation and may impair port infrastructure, raising maintenance
costs (Mehta et al., 2014). As many estuaries are heavily used waterways, they are increasingly modified to maintain navigable
channels and access to ports, altering both their sediment dynamics and hydrodynamics. These alterations can lead to shifts in
morphodynamics, with sediment redistribution potentially causing increased erosion or new deposition zones (Zhou et al.
2019). Moreover, increased turbidity in coastal and estuarine environments has severe negative ecological impacts. It limits
light penetration, reducing primary production and disrupting the food chain by negatively affecting the zooplankton
community and higher organisms that depend on it (De Jonge and Schückel, 2019; Azhikodan and Yokoyama, 2016).
Additionally, with high turbidity and longer residence time of mud, oxygen deficits and hypoxic conditions (oxygen < 2 mg
L$^{-1}$) develop and further diminish the abundance or presence not only of resident fish and other macrofauna, but also of
migratory species thus negatively affecting the biodiversity of these ecosystems (Hoitink et al., 2020; Schulz et al., 2022; Talke
et al., 2009a; Mudge et al., 2007).

Definitions and classifications of layers with high sediment concentrations (Tab. 1) range from mobile suspension, over
(mobile) fluid mud to stationary, consolidated or settled mud, depending on sediment concentrations or based on additional
properties, such as viscosity and/or rheologic behaviour (Kirby and Parker, 1983; Papenmeier et al., 2012; Manning et al.,
2010). Similarly, definitions and characterisations of fluid mud and density gradients vary strongly depending on study area
and research question but such differentiation is typically based on sedimentological or rheological properties (Azhikodan and
Yokoyama, 2018; Manning et al., 2010; Schulz et al., 2022; Winterwerp et al., 2017). The occurrence of fluid mud is
characterized by a sharp density gradient, termed lutocline. In general, the water column is separated from the fluid mud layer
by a single lutocline, depending on the fluid mud characteristics the formation of multiple lutoclines might also be observed
(Ross and Mehta, 1989). The maximum extent of the fluid mud layer ranges from less than 10 % up to 50 % of the water
column (Azhikodan and Yokoyama, 2018; Manning et al., 2010; Ross and Mehta, 1989). The most common lower suspended
particulate matter (SPM) threshold for fluid mud formation is 10 g L$^{-1}$, whereas the upper threshold where the mud layer is





rather classified as stationary than fluid ranges mostly between 100 g L⁻¹ and 480 g L⁻¹ (Papenmeier et al., 2012; Winterwerp et al., 2017; Inglis and Allen, 1957). Based on these definitions the occurrence of fluid mud is reported for many estuaries worldwide (Tab. 1) as well as in many European ports (Shakeel et al. 2022).

**Table 1: Overview of studies that classify fluid mud and/or lower and upper boundaries of high sediment concentrations in estuaries worldwide. The majority of classifications is based on suspended particulate matter or density and not rheological properties. Order is arranged chronologically after publication date (oldest – newest). n.d. = not determined. SSC = Suspended Sediment Concentration.**

| Study | SSC [g L⁻¹] lower threshold | SSC [g L⁻¹] upper threshold | Classification (properties) | River (country) |
|---|---|---|---|---|
| Inglis and Allen (1957) | 10 | 480 | Fluid mud | Thames (Great Britain) |
| Kirby and Parker (1983) | <0.1 | 200 | Stationary suspension (pseudo-plastic) | Severn (Great Britain) |
| | <1 | >150 | Mobile suspension (Newtonian–pseudo-plastic) | |
| Nichols (1984) | n.d. | <3 | Concentrated suspension | James (USA) |
| | 3 | 320 | Stationary suspension | |
| | >320 | n.d. | Settled mud | |
| Wolanski et al. (1988) | >10 | n.d. | Fluid mud | South Alligator River (Australia) |
| Mehta (1991) | >4.4 | 90 | Fluid mud (Mobile Hyperpycnal Layer) | Lake Okeechobee (USA) |
| | 90 | 130 | Stationary Hyperpycnal Layer (Bingham-plastic) | |
| | >130 | n.d. | Cohesive Bed | |
| Kineke et al. (1996) | n.d. | <10 | Mobile suspension | Amazon (Brazil) |
| | 10 | 100 | Mixed fluid mud | |
| | 100 | 330 | Stratified fluid mud | |
| Abril et al. (2000) | 50 | 140 | Fluid mud | Gironde (France) |
| | 140 | 250 | Fluid mud (denitrification layer) | |
| | >250 | n.d. | Fluid mud (Mn(IV)-reduction layer) | |
| Abril et al. (2004) | 10 | 100 | Fluid mud | Loire (France) |
| Uncles et al. (2006) | 10 | 100 | Fluid mud | Humber (Great Britain) |
| | 15 | 40 | Mobile mud suspension | |
| | >40 | >90 | Stationary suspension | |
| Schettini et al. (2009) | >10 | >100 | fluid mud | Tijucas (Brazil) |



| Manning et al. (2010) | 10 | 20 | Mobile suspension | Severn (Great Britain) |
|---|---|---|---|---|
| | 20 | 80 | Settled mud/fluid mud (pseudo-plastic) | |
| | 80 | 220 | Settled mud/fluid mud (Bingham-plastic) | |
| Papenmeier et al. (2012) | 0 | 20 | Sediment suspension | Weser, Ems (Germany) |
| | 20 | 200 | Fluid mud (low viscosity) | |
| | 200 | 500 | Fluid mud (high viscosity) | |
| Azhikodan and Yokoyama (2018) | 3 | n.d. | Fluid mud (based on viscosity >10 Pa s) | Chikugo (Japan) |
| Becker et al. (2018) | >10 | n.d. | Fluid mud | Ems (Germany) |
| Tu et al. (2019) | >15 | n.d. | fluid mud (no lutocline) | Qiantang (China) |
| | >80 | n.d. | fluid mud (based on reduction of turbulence) | |
| Oberrecht (2021) | 10 | 100 | Fluid mud (hindered settling) | Weser, Ems (Germany) |
| | 100 | 250 | Fluid mud (consolidated) | |
| | >250 | n.d. | Consolidated bed | |
| Li et al. (2023) | >10 | n.d. | fluid mud | Jiaojiang (China) |
| Wu et al. (2023) | >10 | >100 | fluid mud | Yangtze (China) |

One of the main drivers for fluid mud formation is the onset of hindered settling at ~10 g L$^{-1}$ where, dependent on floc properties, the floc settling velocity decreases and turbulence is damped (Ross and Mehta, 1989; Dijkstra et al., 2018;
Oberrecht, 2021). Furthermore, fluid mud is characterised as a non-Newtonian fluid (Manning et al., 2010; Oberrecht, 2021) with thixotropic properties (Chanson et al., 2011; Wolanski et al., 1992). Additionally, vertical salinity gradients are associated with the occurrence of fluid mud in multiple estuaries (Haas, 1977; Talke et al., 2009a; Wu et al., 2023).

The temporal variability of fluid mud dynamics has been reported on a short-term timescale such as on a semidiurnal basis in the Ems (Schrottke et al., 2006; Becker et al., 2018) and in many other estuaries worldwide (Becker, 2011; Shakeel et al.,
2022; Wu et al., 2023; Dijkstra et al., 2025). Over a tidal cycle, Becker et al. (2018) distinguished five stages governing the formation and breakdown of fluid mud. Stage 1 is characterized by rapid entrainment at the onset of the flood tide, followed by a reformation of the fluid mud layer within approximately 0.5 hours towards high water slack (HWS) (stage 2). Distinct stratification was observed about 1.5 hours before and after HWS (stage 3). During the ebb phase (stage 4), increased turbulence and shear at the lutocline coincided with a fluid mud share of about 50% across the total depth. This turbulence led
to a reduction in stratification towards low water slack (stage 5), after which the tidal cycle was observed to recommence with the entrainment of the fluid mud layer.





Spring-neap tidal differences have also been investigated for many other estuaries. The majority of the studies concludes that during spring tides, stronger currents lead to more frequent formation and breakdown of the fluid mud layer, along with full entrainment of the fluid mud layer, due to increased turbulence and reduced stability. Whereas, during neap tides the fluid mud layer is often not fully entrained due to decelerating tidal conditions. Thereby, weaker hydrodynamics are associated with more stable conditions, favouring settling and consolidation of suspended matter into a persistent fluid mud layer over multiple tidal cycles (Wu et al., 2023; Abril et al., 2004; Dijkstra et al., 2025; Haas, 1977).

On a seasonal scale, fluid mud occurrence in the Lower Ems River is inversely related to river discharge at Papenburg with a maximum in suspended particle matter (SPM) at the minimum discharge during summer and vice versa (Winterwerp et al., 2017). Thus, at Papenburg, fluid mud occurs mostly from spring to autumn while it is absent over the winter months. Additionally, fluid mud occurrence in summer is linked to the increased occurrence and quality of organic matter, as the organic matter enhances particle-particle interactions, flocculation, thus increasing settling velocities (Lee et al., 2019; Zander et al., 2022b; Gebert and Zander, 2024).

Despite many studies investigating characteristics and/or properties of fluid mud, monitoring and quantifying processes of fluid mud in estuarine environments are still challenging, particular in view of the rapid changes and the varying temporal scales of fluid mud occurrence. Observational data acquisition strongly depends on tidal dynamics as measurements during spring, neap, and intermediate tides yield strong differences (Dijkstra et al., 2025; Manning and Dyer, 2002; Webb and D'Elia, 1980). While (numerical) models were significantly improved to reproduce trends and patterns of the estuarine (sediment) dynamics (Dijkstra et al., 2019b; Nakagawa et al., 2012), they often cover longer time scales (e.g., seasonal or spring-neap variability) of fluid mud formation and break-up. To better characterize processes of fluid mud dynamics and, on the long run, help mitigate issues concerning navigability and occurrence of anoxic conditions due to high sediment concentrations in estuaries, there is still a clear need for high temporal and vertical resolution observational data.

This study aims to examine the temporal and vertical variability of fluid mud dynamics in the Lower Ems estuary, with particular focus on semidiurnal and spring–neap tidal cycles. It further seeks to evaluate the effects of fluid mud occurrence on estuarine hydrodynamics, sediment transport, and more generally on navigation and ecological functioning. Finally, the study aims to establish the importance of integrated long-term and spatially explicit observations in advancing process-level understanding and supporting improved management strategies of hyper-turbid estuaries.

## 2 Study sites

The River Ems, located in NW Germany, has a catchment area of 17,934 km² (Krebs and Weilbeer, 2008) and shares a direct border with the Netherlands. In this study, we focus on the tidal zone of the Ems, which extends landwards until the tidal weir at Herbrum (Ems-km -13.2; Fig. 1), i.e., 13.2 km upstream of Papenburg, where the Lower Ems begins (Ems-km 0.0). The Lower Ems River transitions over the Emden Fairway into the Outer Ems at km 67.76; from that point outwards, the river is classified as sea waterway passing the Ems Estuary and entering the North Sea. From Herbrum to Papenburg, river width



doubles from approximately 60 m to approximately 120 m; at the river mouth the channel width reaches approximately 600 m

(Winterwerp et al., 2017). Upstream discharge is measured at gauge Versen, situated 40 km upstream Herbrum tidal weir and ranges from 20 – 400 m³ s$^{-1}$ with a mean annual discharge of 80 m³ s$^{-1}$ (Talke et al., 2009b; NLWKN, 2017). Average discharge shows strong variations between summer (45 m³ s$^{-1}$) and winter (114 m³ s$^{-1}$) with highest discharges between January and April (Krebs and Weilbeer, 2008; NLWKN, 2017 ).

As a federal waterway, the Ems estuary is a heavily modified river mainly for the purpose of navigability and transport (e.g.,

cruise vessel transit from Papenburg to the harbours of Eemshaven and Emden and the North Sea). The largest anthropogenic interventions include extensive channel deepening and channelization measures as well as tidal control at the Ems barrage in Gandersum (Ems-km 32.2), which is utilised for storm surge control and ship transits (Oberrecht and Wurpts, 2014). These anthropogenic alterations have led to significant morphological and hydrodynamic changes, amplifying tidal asymmetry and tidal range with strong flood tide dominance (Talke and Jay, 2020; Wünsche et al., 2024) and subsequently the accumulation

of fine-sediments within the Ems estuary (Pein et al., 2014; Van Maren et al., 2023). Tidal cycles in the Lower Ems river are characterised by a mean flood time (time between LWS and HWS) between 5 – 5.5 h, and mean time of the ebb tide (time between HWS and LWS) with approximately 7 h (Winterwerp et al., 2017). The tidal asymmetry with strong flood dominance has led to a distinct regime shift in the Ems estuary from low to high sediment concentrations and hyper turbidity (Dijkstra et al., 2019; Winterwerp and Wang, 2013; Borgsmüller et al., 2016). Especially during dry periods along with low upstream river

discharge, the sediment concentrations range from <1 g L$^{-1}$ at the surface to several 100 g L$^{-1}$ near the bottom (Becker et al., 2018; Papenmeier et al., 2012; Talke et al., 2009b). The continuous accumulation of fine cohesive particles and flocs initialises the formation of fluid mud, altering the estuarine hydrodynamics further (Becker et al., 2018; Dijkstra et al., 2019; Borgsmüller et al., 2016). During periods of increased river discharge, the fluid mud layer thins, leading to compaction of the mud bed and/or enrichment in coarse grains (e.g., sand), which results in armouring of the river bed (Winterwerp et al., 2017). Maximum

turbidity values are observed at Papenburg (Ems-km 0.0), rather than where the estuarine turbidity maxima (ETM) based on the strongest salinity gradient is reported, i.e., between Pogum (Ems-km 33.6) and Terborg (km 24.5), due to continuous tidal pumping (Dijkstra et al., 2019b; Oberrecht, 2021; Van Maren et al., 2023). The ETM is characterized by comparably low salinity which ranges from 3 PSU at Terborg to 0.5 PSU at Weener (Krebs and Weilbeer, 2008; Talke et al., 2009a).

Two complementary field campaigns, a temporary mooring deployment and a series of dedicated measurement campaigns,

were conducted to collect hydrodynamic and sediment data. The temporary mooring was deployed near Weekeborg (Ems-km 11.8) while dedicated measurement campaigns were conducted near Coldam (Ems-km 13.2). The measurement sites were chosen based on the availability of dolphins for anchoring and logistical proximity to the overnight mooring pontoon at the mouth of the River Leda (Fig. 1).





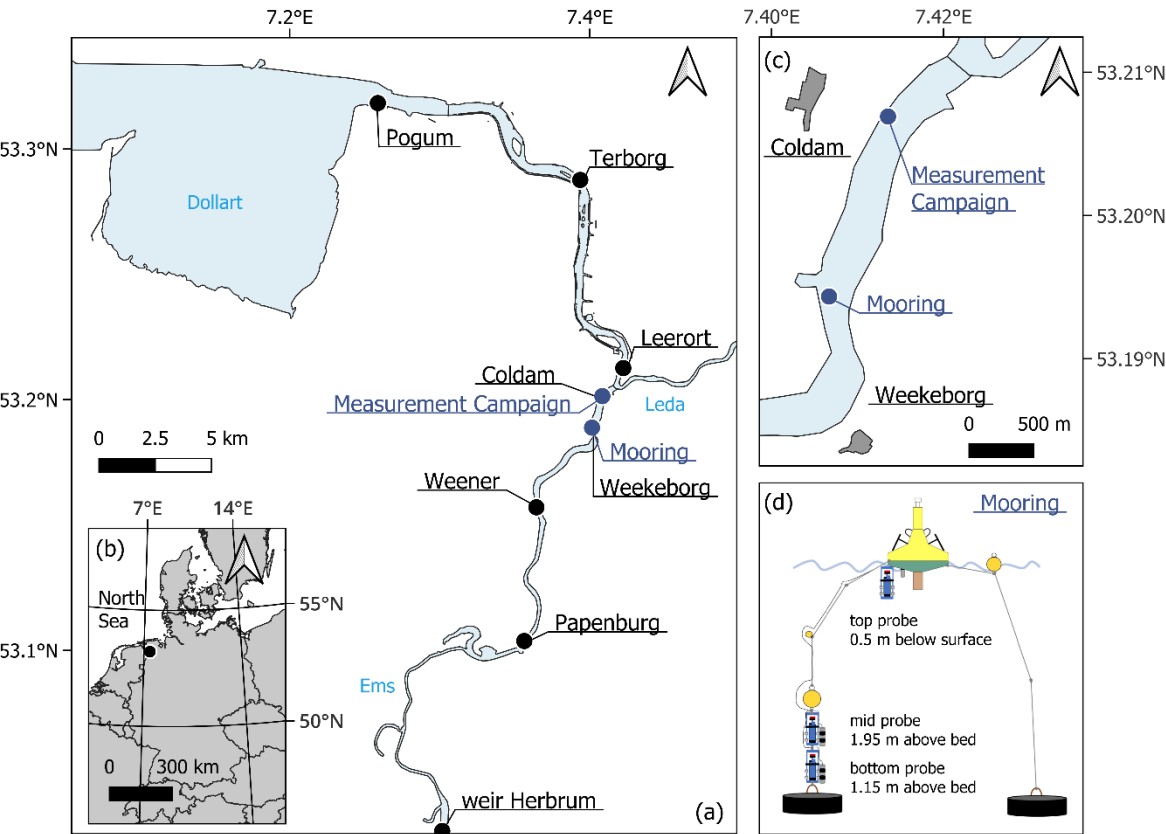

**Figure 1: Ems Estuary and measurement setup. (a) Location of the study area with temporary moooring deployment in 2019 near Weekeborg (Ems-km 11.8) and measurement campaigns in 2023 near Coldam (Ems-km 13.2). Black dots show locations of continuous total suspended solids stations from the NLWKN (no monitoring station at Herbrum). (b) Location of the study area in NW Germany at the German-Dutch border. (c) Location of the temporary mooring and the measurement campaigns near settlements Coldam and Weekeborg, both located at the left side of the river. (d) Setup of temporary mooring deployment used in 2019 shows location of bottom, mid, and top probe. Grey probes are total suspended solid sensors, blue probes are multiparameter sensors.**

## 3 Methods and data

### 3.1 Observational data and analysis of vertical profiles

To analyse spring neap tidal dynamics, two measurement campaigns were conducted in September 2023. Measurements were carried out with the R/V *Friesland* near Coldam (Ems-km 13.2; Fig. 1). Each campaign comprised two measurement days, covering two days of spring tide and two days of neap tide. Measurements were performed over approximately 8 h per day, beginning about 3 h before LWS and ending about 5 h after, thereby capturing the decreasing limb of the ebb phase, ebb low water, and the majority of the subsequent flood tide. Vertical profiles of in situ density were recorded using a tuning fork



density probe (RheoTune, Stema systems) and a SeaGuard Recording Current Meter (RCM) multi-parameter probe (Xylem/ Aanderaa Data Instruments AS, Bergen, Norway), the latter mounted on top of the density probe. Vertical profiles were obtained every 10 min using a sideboard winch, except during slack tide; when the vessel was realigned with the flow direction after reversal. Acquisition of a single profile typically required less than 1 min, depending on winch and current velocity as well as water depth. The RheoTune recorded at 20 Hz, while the multi-parameter probe sampled at 0.5 Hz, resulting in ~1000 and 10–20 data points per vertical profile, respectively. Consequently, the vertical resolution was <1 cm for the RheoTune and approximately 20–30 cm for the multi-parameter probe. In total, 85 vertical profiles were collected over two days during neap tide and an additional 71 profiles during spring tide.

Vertical profiles were analysed based on in situ density measurements. To allow for direct comparisons of the spring and neap tidal cycles we separated the dataset with regard to observed build-up and break-up of the fluid mud layer before and after LWS and maximum current velocity. Here, we identified five different stages adjusted after Becker et al. (2018) attributed to the following time intervals: 3 h – 0.5 h before LWS (Stage I), 0.5 h before to 0.5 h after LWS (Stage II), 0.5 h – 2 h after LWS (Stage III), 2 h – 3.5 h after LWS (Stage IV), and 3.5 h – 5 h after LWS (Stage V). For Stage I, when the mobile mud layer reached its maximum height, the impact of the fluid mud layer on the flow velocity was evaluated. The height of the lutocline $h_L$ was defined as the depth where total suspended solids exceeded 10 g L$^{-1}$. Depths were normalised to the maximum water depth of the vertical profile (z), such that the bed corresponds to 0 and the water surface to 1. The mean flow velocity above the lutocline ($v_{flow}$) was computed from vertical profiles, assuming that the fluid mud layer below does not contribute to flow. The corresponding unit discharge was calculated as $q = v_{flow} h_{flow}$.with $h_{flow} = z - h_L$. To explore the effect of varying fluid mud thickness, the active flow layer ($h_{flow}$) was adjusted to fractions $f = 0.2, 0.4, 0.6, 0.8$ of the normalised water column while keeping $q$ constant, yielding the adjusted mean velocities $v_{flow}(f) = q/h_{flow}(f) = q/(fz)$. This represents a simplified 1D approach, assuming uniform flow across the cross-section and neglecting horizontal variations or turbulence within the flow layer.

The analysis of the five stages focused on the stability of the fluid mud layer and its disintegration. In case of a local density gradient a boundary shear layer develops under hydrodynamic pressure. The stability of the individual layers was quantified with the dimensionless gradient Richardson number (Ri$_g$), which describes the relationship between vertical stratification stability and vertical shear (Becker, 2011). We calculated Ri$_g$ for every depth (h, [m]) below the surface with the following equation (Eq. 1):

$$Ri_g(h) = -(h_L - z) \cdot g(\rho_h - \rho_L)/\rho_h \cdot 1/(\overline{u_h} - \overline{u_L})^2 \tag{1}$$

here, $h_L$ [m] is the lower boundary depth of the lutocline (L). z [m] is the maximum water depth of the vertical profile, g [m s$^{-1}$] is the gravity acceleration, $\rho_h$ [kg m$^3$] and $\rho_L$ [kg m$^3$] are the density at each depth and at the lutocline ($h_L$). $u_h$ and $u_L$ are the time-averaged velocities at depth h and at $h_L$ respectively. The critical Ri$_g$ (0.25) indicates either favourable conditions for the damping (> 0.25) of turbulence, indicating stable stratification, or the onset (< 0.25) of turbulence and mixing where stratification becomes unstable (Desaubies and Gregg, 1981; Fernando, 1991; Macdonald and Horner-Devine, 2008).



Further, we quantified the damping at the lutocline with the buoyancy frequency (N, [s⁻¹]) which represents the frequency of an oscillation with which a fluid particle is transported vertically (Kundu et al., 2015). Here, increased N values (> 0.4) are associated with stratified conditions and indicate decoupling between the mud layer and the water column. Low N values (~ 0.2), however, represent stronger vertical movement and mixing of the particles (Becker et al., 2018). The buoyancy frequency for each depth is given by (Eq. 2)

$$N(h) = \sqrt{\frac{g}{\rho_h} \left| \frac{(\rho_h - \rho_L)}{(h - h_L)} \right|} \tag{2}$$

We further quantified the vertical dispersion by the entrainment rate ($u_e$). Neglecting viscous effects (Kranenburg and Winterwerp, 1997), the entrainment rate is given by (Eq. 3)

$$u_e(h) = u_* \sqrt{0.5/(5.6 + Ri_b)} \tag{3}$$

The entrainment rate depends on the shear velocity ($u_*$, [cm s⁻¹]) which varies greatly between slack water and maximum flow velocity for each tide. The shear velocity is approximated with reference to the lutocline by (Eq. 4)

$$u_* = \kappa \cdot z \cdot |(u_h - u_L)/(h_L - h)| \tag{4}$$

Additionally, for the calculation of the entrainment rate we used the bulk Richardson number ($Ri_b$) given by (Eq. 5)

$$Ri_b = z \cdot \frac{\Delta b}{u_*} \tag{5}$$

with the buoyancy difference $\Delta b$ (Eq. 6)

$$\Delta b_h = g \, |(\rho_h - \rho_L)| \, / \, \rho_h \tag{6}$$

Thereby, we were able to quantify stratification, vertical dispersion, and entrainment of the fluid mud layer and its interaction with the water column across ebb and flood and the spring and neap tidal cycles.

**3.2 Observational data and analysis of temporary mooring**

The temporary mooring was installed between August and September 2019 (total time of 6 weeks), near Weekeborg (Ems-km 11.8), between Weener and Leerort (Fig. 1). The location was selected due to accessibility for maintenance, and under consideration of local bathymetry and navigability requirements, such as no interference with the fairway and ship traffic. Our mooring system consisted of a set of instrument pairs, installed in three different depths along a steel cable attached to a buoy (Fig. 1). The first sensor pair was directly mounted to the buoy and submerged approximately 0.5 m below water surface. The second sensor pair was linked with a steel cable to a smaller submerged floating marker and was located at a fixed height of approximately 1.95 m above bed. The third sensor pair was directly mounted above the anchor stone approximately 1.15 m above the river bed (Fig. 1). Each pair included a self-sustaining RCM multi-parameter probe (identical to the one used in the measurement campaigns) and a total suspended solids (TSS) probe (Solitax hs-line sc, Hach Lange GmbH). The RCM was used for the collection of key physical parameter such as temperature, pressure, conductivity, dissolved oxygen, current velocity, and turbidity. The additional TSS probe was necessary to ensure reliable TSS measurements within the ETM. The TSS probe registers total suspended solids between 0.1 – 500 g L⁻¹ reliably with an accuracy of ± 1 % (Hach Lange GmbH,



2022), thus complementing the default equipment (RCM probes) whose upper detection limit is 2000 FTU (~ 2 g L$^{-1}$) (Aanderaa Data Instruments AS, 2020 ). The TSS probes were equipped with an automatic wiper to minimize biofouling. As

the TSS probes rely on direct power supply via cable, solar panels were mounted to the buoy and connected to a battery to ensure power supply over the deployment period. Remote data transmission was achieved with a netDL Data Logger (OTT HydroMet). To reduce lateral motion of the vertical measuring transect a second anchor stone on the opposite site of the surface buoy provides stability and prevents extensive twisting during tide reversal (Fig. 1).

Time series analysis of the mooring data was carried out focusing on three different objectives, i) the occurrence of fluid mud

(i.e., time of fluid mud presence), ii) the height of fluid mud, iii) driving forces/processes connected to build-up and break-up of fluid mud. Regarding i) the occurrence of fluid mud, we calculated the time fraction of fluid mud occurrence for each tidal cycle to increase comparability and account for tidal asymmetry between ebb and flood. The time fraction was based on the cumulative time of fluid mud occurrence (bottom TSS record exceeding 10 g L$^{-1}$) and the sampling interval (5 min) divided by the total time of each tide. Based on results of i) and ii) we conducted a more in-depth time series analysis with focus on

four different states of fluid mud occurrence. Here, we differentiated between: absence, persistent, break-up, and build-up. For each state we investigated changes in the vertical salinity gradient (with $\Delta S = S_{top}-S_{bottom}$), dissolved oxygen, current velocity, and total suspended solids. To increase comparability, each state was further investigated at the time scales of a single semidiurnal cycle, by analysing a representative 24 h period comprising ebb tide and two LWS scenarios and the subsequent flood tides.

## 4 Results and discussion

### 4.1 Vertical dynamics of fluid mud occurrence over a semidiurnal cycle driven by spring and neap tide

Our measurement campaigns confirmed the near-continuous presence of fluid mud throughout the semidiurnal tidal cycle, with only a brief absence (<1 h) around LWS, regardless of tidal conditions (Fig. 2). The maximum vertical extent of fluid mud occurred during the ebb tide, covering 50–60% of the water column near LWS, whereas during flood tide it comprised

only 30–40%. Based on the temporal variability of the fluid mud layer, we identified five characteristic stages within the semidiurnal cycle (Fig. 2). Stage I corresponds to the persistent presence of fluid mud during the ebb tide. Stage II marks rapid thinning of the layer shortly after LWS, concurrent with a sharp increase in current velocity. Stage III represents the temporary absence of the fluid mud layer approximately 1.5 h after LWS, as current velocities gradually decrease. Stage IV corresponds to the renewed thickening of the fluid mud layer during the flood tide, whereas Stage V represents its persistence as current

velocities decline toward HWS, marking the transition to reduced hydrodynamic forcing. The consistent occurrence of these five stages under both spring and neap tides highlights the robust tidal control on the formation, persistence, and absence of fluid mud (Wu et al., 2022).



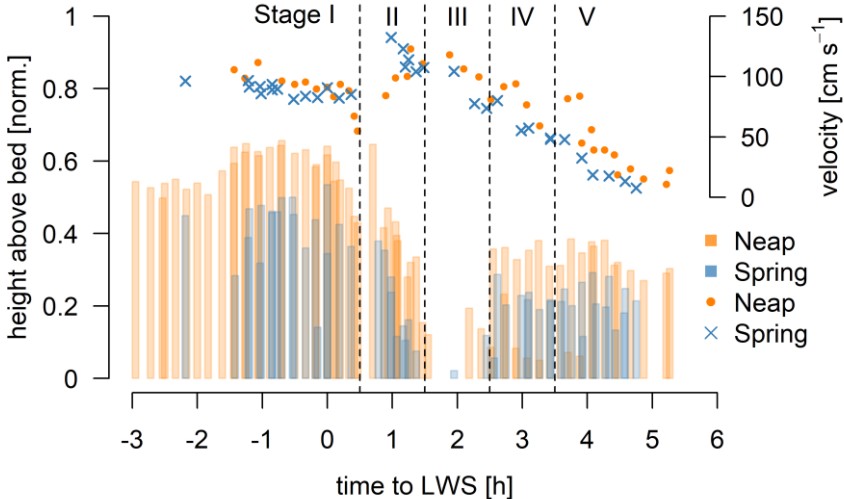

**Figure 2: Lutocline height above bed for spring (blue) and neap (orange) tide with mean flow velocity measured. Depth normalised based on height above bed. Data shown from four measurement days in 2023. Stages I – V are separated by black vertical dashed lines.**

Strong differences between spring and neap tides were observed in terms of fluid mud height (normalised depth) and maximum current velocity (Fig. 2). During the ebb phase of neap tides, flow velocities were comparable to the maximum velocities recorded during ebb-spring tides. However, during the flood phase, neap tides exhibited stronger flow velocities than those observed during spring tides. Here, the total vertical extent of fluid mud was similar during both spring and neap tides (approximately 2 – 2.5 m), but the relative proportion differed substantially. During the spring tide, fluid mud accounted for approximately 40% of the water column, whereas during the neap tide, it reached approximately 60%. This indicates that the relative thickness of the fluid mud layer modulates turbulence and current velocity in the overlying water column (Wu et al., 2023). A key hydrodynamic control underlying this behaviour is the nonlinear effect of lutocline height on flow velocity (Fig. 3). 1D modelled flow velocity (Sec. 3.1) increases notably with increasing lutocline height, primarily due to the reduction of the hydrodynamic cross-section. This relationship is supported by measured velocities approaching 80 cm s$^{-1}$ (Fig. 3).




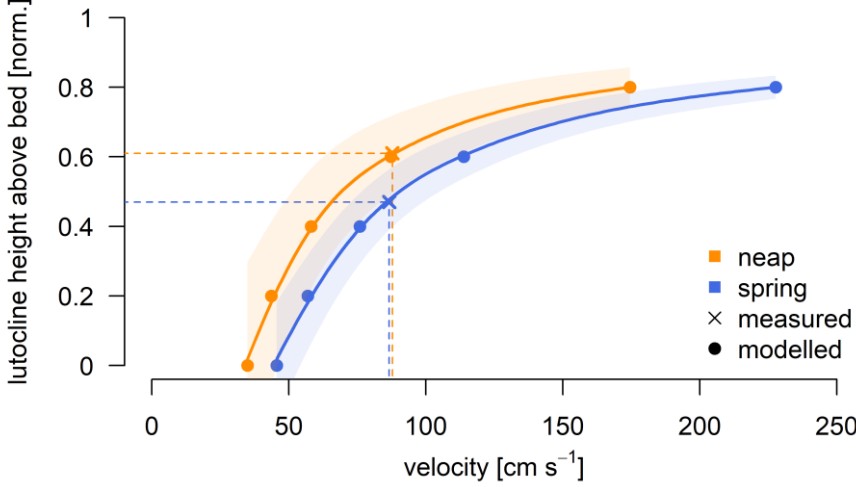


**Figure 3: Schematic impact of the lutocline on flow velocity separated by spring and neap tide. 1D-modelled data is based on mean velocity over the lutocline at Stage I, when lutocline reaches its maximum extent. Increase in flow velocity with increased lutocline height above bed is observed, as hydrodynamic cross-section is reduced.**

In terms of hydrodynamics, the average entrainment rate was distinctly higher during the end of the ebb tide (Stage I) than during the flood tide (Stages III–V) (Fig. 4). Following LWS, and particularly under flood-spring tide conditions (Stages III–V), buoyancy frequency ($N^2$) increased markedly as the fluid mud layer thinned, reflecting the development of stronger vertical density stratification. During ebb-neap tides, however, buoyancy frequency remained nearly constant (Tab. 2). The observed changes in $N^2$ and $Ri_g$ indicate that semidiurnal tidal forcing regulates the balance between stratification and turbulence,

controlling vertical mixing and stability of the fluid mud layer (Becker et al., 2018; Wu et al., 2023). As current velocities decreased, the gradient Richardson number ($Ri_g$) became very low and the entrainment rate dropped substantially. This decline in entrainment coincided with the rise in buoyancy frequency, indicating enhanced stratification and suppressed vertical mixing (Fig. 4).

Further, differences in fluid mud density were observed before LWS (Stage I), when a second, denser fluid mud layer ($C_s =$

$50 – 100$ g $L^{-1}$) was detected near the bed (Fig. 4a). Fluid mud with concentrations exceeding 50 g $L^{-1}$ are generally considered quasi-stationary, as its persistence enhances particle flocculation and settling processes (Oberrecht, 2021; Wu et al., 2023). During ebb-neap tides, this dense layer was noticeably thicker and more stable, accompanied by a stronger vertical density gradient in the water column (Fig. 4a, Fig. 4b). This is linked to increased $Ri_g$ values (i.e., $Ri_g > 0.25$), indicating dampened turbulence and an increase in buoyancy frequency, particularly above the lutocline (Tab. 2). Below the lutocline, buoyancy

frequency also increased, suggesting limited vertical exchange and the development of internal shear. In contrast, during ebb-spring tide, the fluid mud layer was less dense and $Ri_g$ values were lower ($< 0.25$), consistent with more turbulent conditions that enhanced mixing and dispersion of the layer during Stage II (Fig. 4b).



We argue that the reduced relative height of the fluid mud layer during spring tides (Fig. 3) contributes to lower turbulence at the lutocline. This occurs because a higher water column above the lutocline distributes momentum more evenly, thereby

decreasing shear stress at the top of the fluid mud layer (Chmiel et al., 2021). Our observations (Fig. 4, Tab. 2), together with previously published results (Becker et al., 2018), demonstrate lower entrainment rates during spring tides compared to neap tides. During the flood phase of spring tide, when current velocities peak, increased tidal amplitude and water depth further dampen turbulence at the lutocline, enhancing vertical stratification. These conditions correspond to increased buoyancy frequencies and reduced vertical mixing (Tab. 2). Such results are consistent with Tu et al. (2019), who reported decreased

turbulence and entrainment under spring tidal conditions. The associated reduction in turbulence facilitates increase of floc settling and subsequent consolidation of the fluid mud during spring tides (Fig. 4) (Wolanski et al., 1992; Xu et al., 2020; Wu et al., 2023). In contrast, during flood-neap tide, a thicker fluid mud layer is observed, resulting in higher flow velocities in the water column above (Fig. 2). The associated increase in turbulence disrupts stratification and promotes vertical mixing, driving increased entrainment compared to flood-spring tide than during flood-neap tide (Fig. 4). Accordingly, shear flow at the

lutocline is also lower during flood-spring tide than during flood-neap tide.

Stage IV was marked by the rapid reappearance of the fluid mud layer, with the denser material fully reworked within 0.5–1 h after LWS as current velocities increased (Fig. 4d). These patterns align with previous observations (Abril et al. (2000), Becker et al. (2018), Winterwerp et al. (2017), Wu et al. (2022, 2023)). During ebb-neap tides, entrainment rates in Stage IV exceeded those observed under ebb-spring conditions (Fig. 4d and Tab. 2), while the fluid mud layer was less stable and

stratification weaker, as indicated by low $Ri_g$ values and higher buoyancy frequencies (Fig. 4d). In Stage V, turbulence decreased (reflected by higher $Ri_g$ values), yet entrainment and vertical mixing persisted during ebb-neap tides, whereas ebb-spring tides remained more stable with reduced mixing (Fig. 4e). These observations reveal a recurring cycle of stability and instability in the fluid mud layer, with alternating periods dominated by turbulence or stratification (Wu et al., 2022). The shifts in $Ri_g$ values and buoyancy frequency across stages highlight how tidal dynamics control fluid mud thickness and

entrainment rates (Becker et al., 2018; Wu et al., 2022). The temporary reduction in the effective flow area by dense fluid mud occurs only briefly during the tidal cycle (Fig. 2), supporting the interpretation of the dense layer as dynamics and fully entrained by Stage III (Fig. 2), in contrast to the static definition reported by Abril et al. (2000).










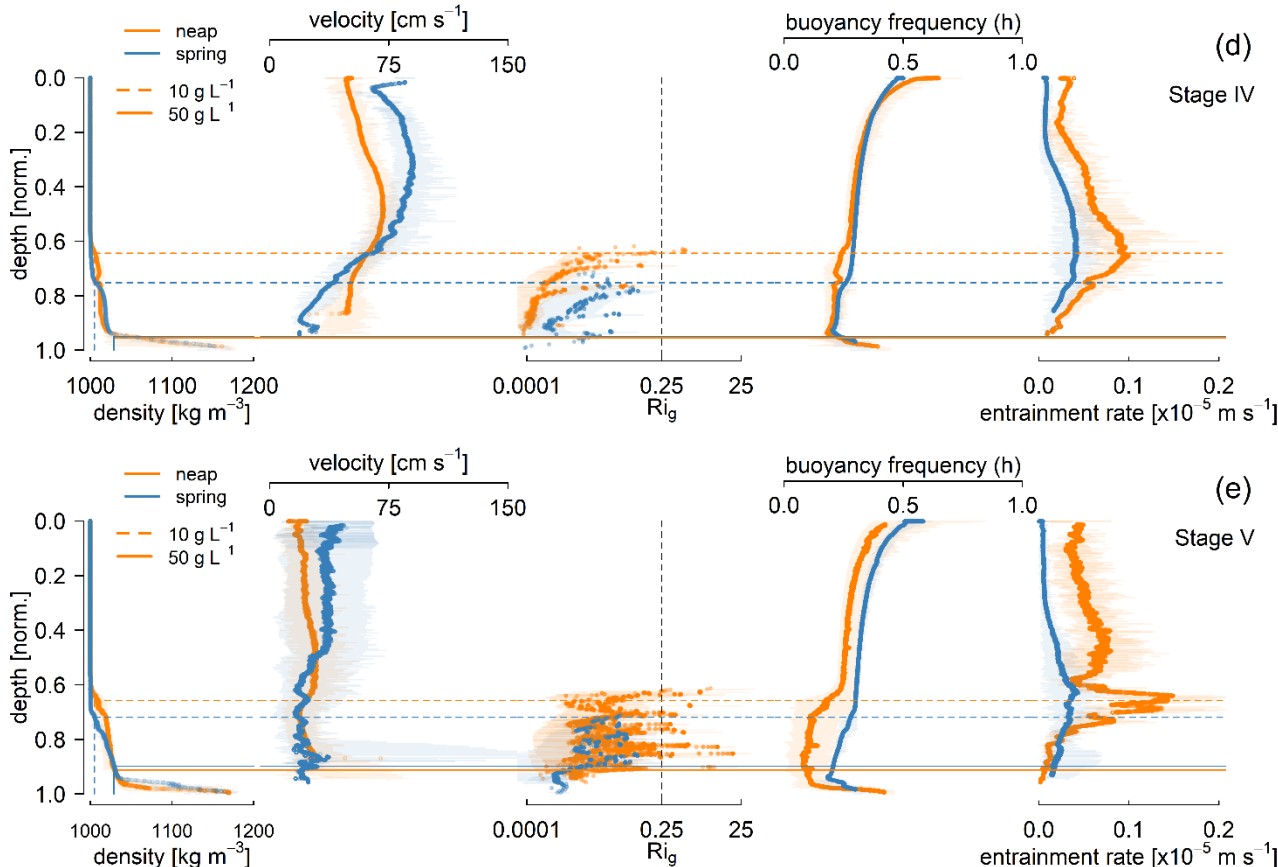

**Figure 4: Density, velocity, Richardson Gradient number ($Ri_g$), buoyancy frequency and entrainment rate for Stages I – V (a – e)**
**over ebb – flood tide. Depth is given as normalised [norm.] depth. Separated to spring (blue) and neap (orange) tide. Position of 10g L$^{-1}$ (dashed) and 50 g L$^{-1}$ (solid) lutoclines shown by horizontal lines. Vertical black line at critical $Ri_g$-value = 0.25, which separates more turbulent (< 0.25) from more stable (>0.25) conditions.**





**Table 2: Quantification of entrainment rate, buoyancy frequency, and location of lutocline for Stages I – V over ebb – flood tide and separated by spring – neap tide. Δt is the time differences to low water slack (LWS), n is the number of profiles. $h_{norm}$ is the normalized height. $Ri_g$ is the gradient Richardson number. Entrainment rate and buoyancy frequency are given as mean value above and below the lutocline.**

| Stage | Δt LWS [h] | | vertical profiles [n] | | Entrainment rate [x10⁻⁵ m s⁻¹] | | | | Buoyancy frequency [s⁻¹] | | | | $h_{norm}$ ($Ri_g >$ 0.25) | |
|---|---|---|---|---|---|---|---|---|---|---|---|---|---|---|
| | from | to | Neap | spring | Neap | Spring | Neap | Spring | Neap | Spring | Neap | Spring | Neap | Spring |
| | | | | | Above lutocline | | Below lutocline | | Above lutocline | | Below lutocline | | | |
| I | -3.0 | 0.5 | 30 | 19 | 7.7 | 6.0 | 7.3 | 6.4 | 0.44 | 0.32 | 0.20 | 0.20 | 0.39 | 0.53 |
| II | 0.5 | 1.5 | 11 | 11 | 4.8 | 7.0 | 8.2 | 8.8 | 0.44 | 0.42 | 0.23 | 0.29 | 0.56 | 0.84 |
| III | 1.5 | 2.5 | 7 | 4 | 2.6 | 3.7 | - | 0.1 | 0.44 | 0.64 | 0.25 | 0.40 | 0.79 | 0.82 |
| IV | 2.5 | 3.5 | 11 | 9 | 5.0 | 2.2 | 5.9 | 2.1 | 0.36 | 0.35 | 0.23 | 0.28 | 0.63 | 0.77 |
| V | 3.5 | 5.5 | 14 | 13 | 5.2 | 1.4 | 5.4 | 2.4 | 0.31 | 0.38 | 0.13 | 0.24 | 0.62 | 0.76 |

## 4.2 Temporal dynamics of fluid mud occurrence on six-week and spring-neap tidal time scale

The analysis of the occurrence of fluid mud, based on presence of fluid mud as fraction of tide length, was performed for ebb and flood tides separately. Based on ebb tides, our analysis revealed recurring cycles of build-up, persistent state, break-up, and no fluid mud occurrence over the six-week monitoring period, covering three full spring-neap cycles (Fig. 5, Fig. 6). We categorised the transition from absence to a persistent state (defined as fluid mud present for more than 50% of the tidal duration) as the build-up phase, typically occurring over approximately two days. The persistent state, when fluid mud was 345 present throughout most of the tidal cycle, lasted more than 1 week. The break-up phase followed, with fluid mud occurrence declining sharply from over 60% to below 20% of the tidal duration within 3–4 ebb tides. During the transition from intermediate to neap tides, the persistent state gave way to complete absence, with no fluid mud observed for 3–4 consecutive tidal cycles. These recurring cycles closely tracked the spring-neap tidal variation, with build-up and persistence occurring during spring tides, and absence during neap tides. These cycles reflect the strong influence of spring-neap tidal modulation, 350 with build-up and persistence generally occurring during higher-amplitude spring tides (Kirby and Parker, 1983). However, local conditions, including shorter tidal duration during flood tides, stronger velocities following LWS, and weak stratification, allow fluid mud to persist even during some neap tides in the study area (Sec. 4.1).

Fluid mud height estimates were limited by the vertical resolution of the mooring probes, which allowed classification into <1.15 m, 1.15–1.95 m, and > 1.95 m above the riverbed (Fig. 1). During the persistent state, fluid mud exceeding 1.95 m was 355 present for approximately 20% of the ebb tide (Fig. 5), whereas during build-up and break-up phases it was observed for less than 10% of the time at this height. However, fluid mud height did not consistently correlate with tidal duration; for example, during one ebb tide with <10% occurrence, concentrations >10 g L⁻¹ were still recorded at the middle probe, suggesting the



layer exceeded 1.95 m and covered more than half of the water column at LWS. This indicates dense fluid mud layers can
form rapidly, independent of the overall tidal duration (Becker et al., 2018).

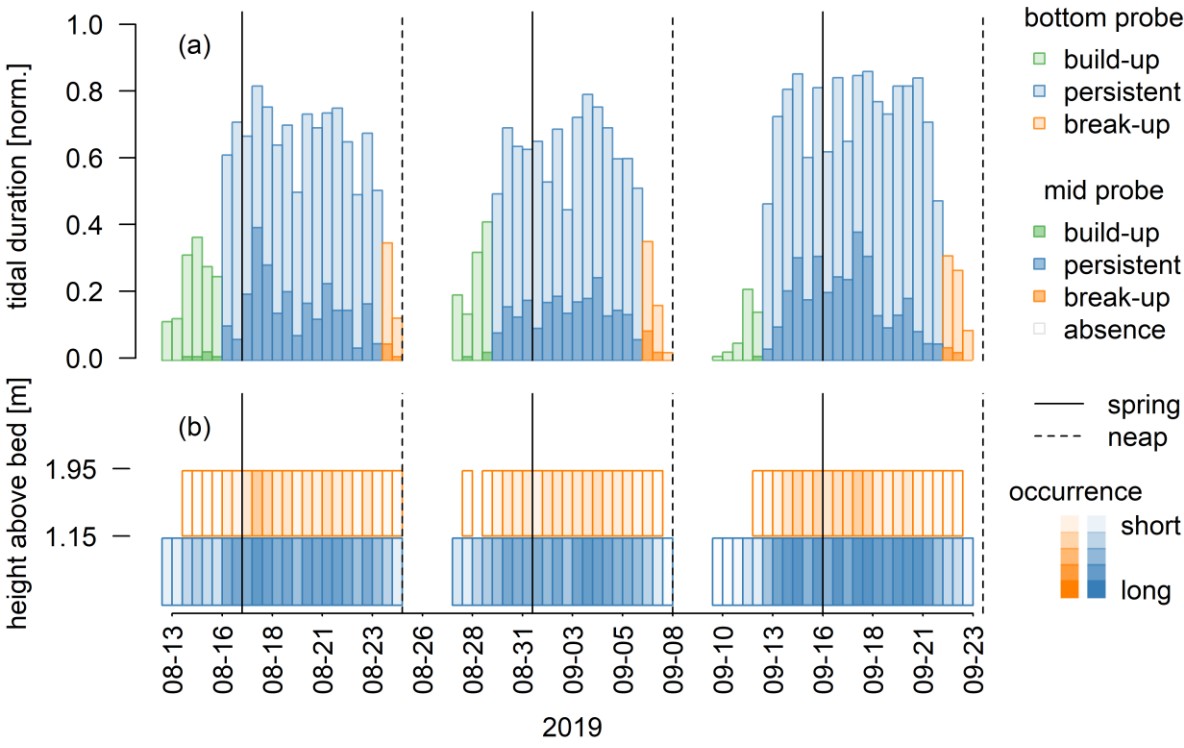

**Figure 5: (a) Presence of fluid mud during ebb tides based on occurrence time per tide length (tide length is normalised [norm.]) for
the mid [dark colours] and bottom [light colours] probe (located 1.15 m and 1.95 m above the bed). Identified stages are based on
normalised tide length. Build-up (green) <0.4, persistent state (green) (>0.4), break-up (orange) (<0.4), absence (white/grey) (<0.1).
(b) Fluid mud occurrence in bottom [blue] and mid [orange] total suspended solid probe based upon concentration > 10g L$^{-1}$. Opacity
of bars (high and low) is normalised based on duration of fluid mud occurrence over a tide (a) (darker colours equal longer fluid
mud occurrence).**





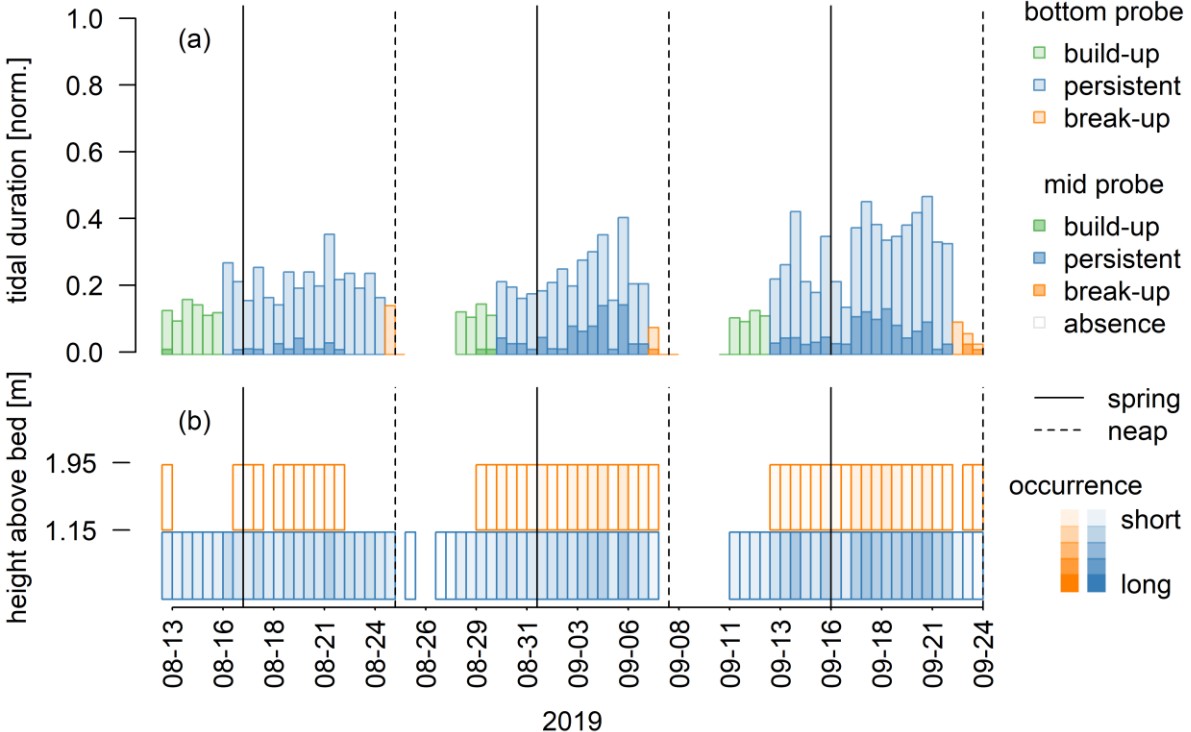

**Figure 6: (a) Presence of fluid mud during flood tides based on occurrence time per tide length (tide length is normalised [norm.]) for the mid [dark colours] and bottom [light colours] probe (located 1.15 m and 1.95 m above the bed). Identified stages are based on Figure 5 (a). (b) Fluid mud occurrence in bottom [blue] and mid [orange] total suspended solid probe based upon concentration > 10g L⁻¹. Opacity of bars is related normalised tide length (darker colours equal longer fluid mud occurrence). Opacity of bars (high and low) is normalised based on duration of fluid mud occurrence over a tide (a) (darker colours equal longer fluid mud occurrence).**

Fluid mud was present for a much smaller fraction of the flood tide (Fig. 6) compared to the ebb tide (Fig. 5). During the build-up phase, fluid mud occurred for less than 20% of the flood tide duration. This increased to around 30% during the persistent state, reaching a maximum of more than 40% during the third cycle (2019-09-17 to 2019-09-22). In contrast to ebb tides, the duration of fluid mud occurrence during flood tides increased over recurring cycles, suggesting a cumulative effect of sediment deposition or lateral transport over multiple tidal cycles (Winterwerp, 2006; Wu et al., 2023). In the second and third cycles, flood tide fluid mud occurrence increased from around 20% near spring tide to approximately 40% around the following intermediate tide, with more than 10% occurrence at the mid TSS probe, compared to less than 10% in the first cycle. Such differences were not observed during ebb tides (Fig. 5). Although the vertical extent of the fluid mud layer was similar during both ebb and flood tides, the ebb tide layers covered the majority of the water column due to tidal amplitude differences, consistent with our measurement campaigns. This highlights the importance of tidal asymmetry: longer ebb tides favour fluid mud accumulation over a greater proportion of the water column, whereas shorter flood tides limit deposition duration (Allen et al., 1977; Dronkers, 1986).



The water level curve (Fig. 7) illustrates the tidal asymmetry, with longer ebb and shorter flood tides. After LWS [time [h] ~ 7) flow velocity increased rapidly, with the highest rates observed at the top probe. These peak velocities (> 100 cm s$^{-1}$) were

390    recorded for less than 2 h. A distinct vertical gradient was present, with higher velocities near the surface and lower velocities near the bed. Towards HWS (time [h] ~ 12) current velocity gradually decreased to the pre-LWS values (~ 50 cm s$^{-1}$) within < 3 h. The entire tidal cycle was characterised by generally low TSS values and absence of fluid mud. Dissolved oxygen concentration followed the tidal water level, decreasing slightly towards LWS and increasing again toward HWS, with no vertical gradient observed. Salinity showed a similar pattern, with ΔS values remaining constant, indicating no vertical

395    stratification. This suggests that during fluid mud absence states, vertical mixing was sufficient to homogenise oxygen and salinity, limiting the formation of stratified fluid mud layers (Schmidt and Malcherek, 2021; Wehr, 2014).

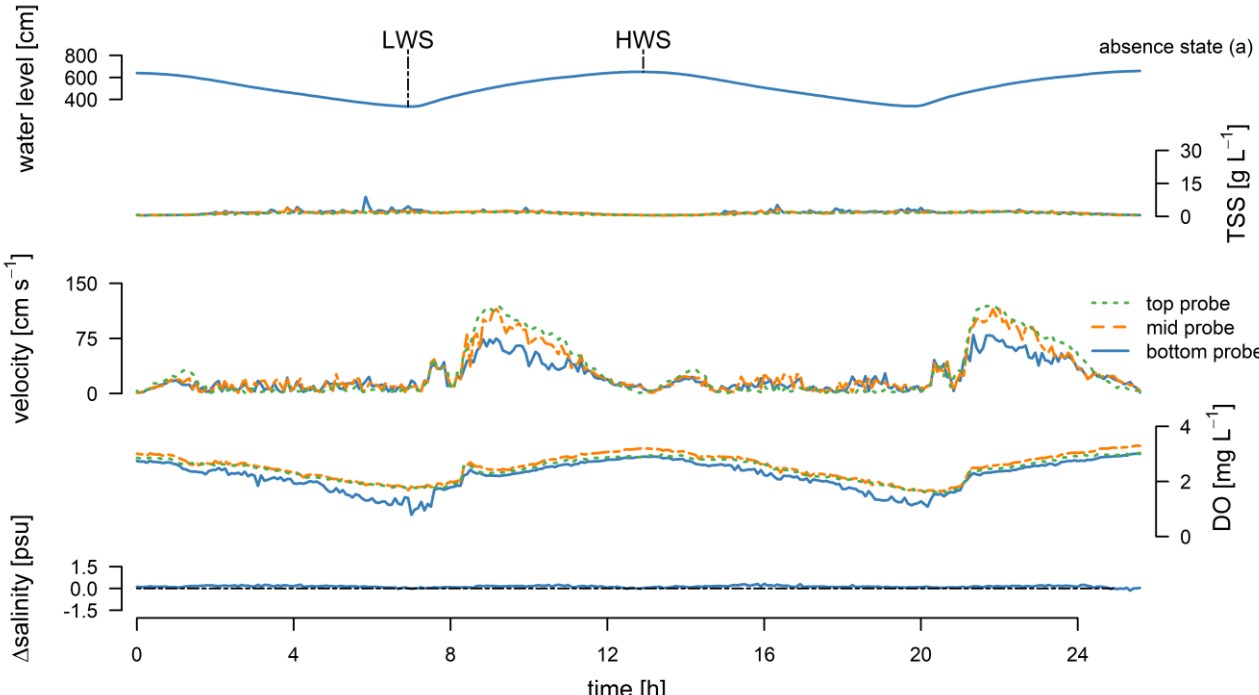











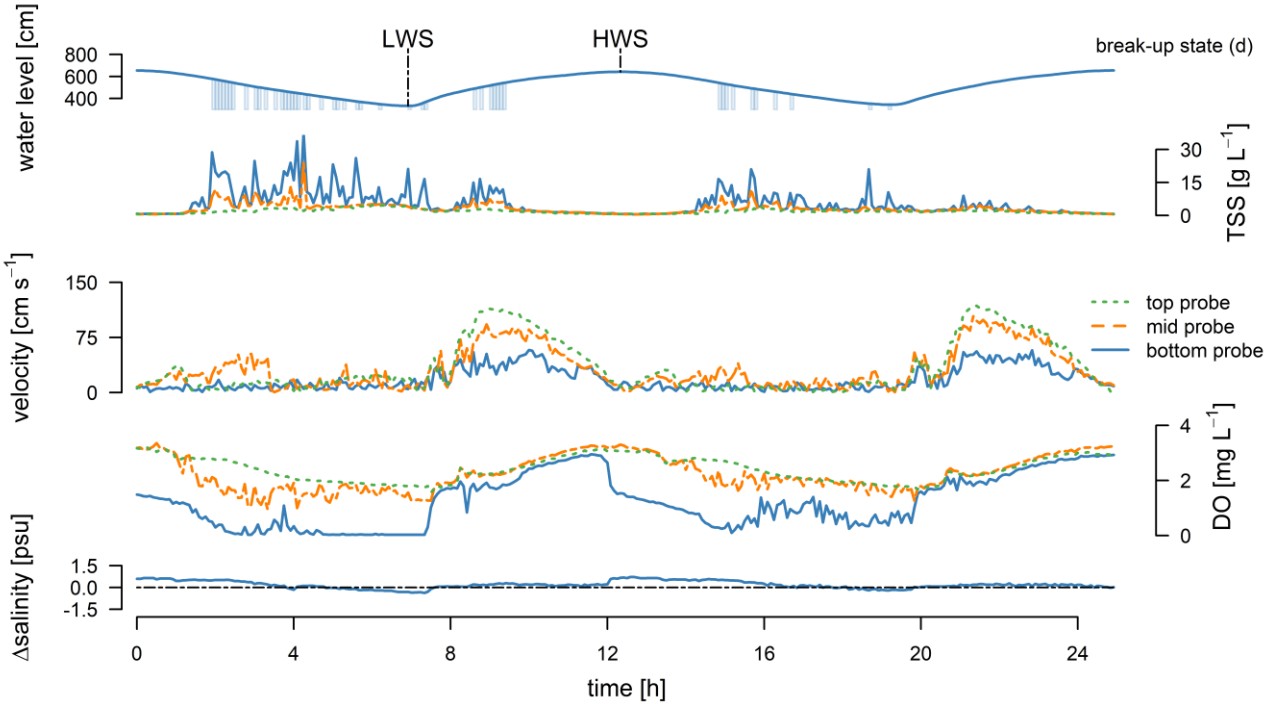

**Figure 7: (a) Water level, total suspended solids (TSS), velocity, oxygen, and Δsalinity (ΔS) during absence state of fluid mud from temporary mooring deployment 2019 near Weekeborg . Parameters recorded over semidiurnal cycle by multi-parameter probe for stages identified from temporary mooring deployment. LWS = low water slack, HWS = high water slack. Vertical bars in water level plot show TSS measurements > 10 g L⁻¹. (a) absence is associated with neap tide, (b) intermediate tide from neap to spring tide, (c) spring tide, and (d) intermediate tide between spring to neap tide. (b) – (d) show semidiurnal cycle for build-up state, persistent state, and break-up state as classified from normalised tide length (Fig. 2(a)).**

The first continuous occurrence of fluid mud was characterized as the build-up phase (Fig. 7b). TSS values gradually increased

from below 10 g L⁻¹ to approximately 30 g L⁻¹ within less than 2 hours recorded by the mid TSS probe, around the second

LWS (time [h] ~ 20), indicating a minimum fluid mud height exceeding 1.15 m. The maximum values were recorded just

before the onset of tidal flow reversal and the increase of flow velocity. A vertical gradient in oxygen concentrations was

observed, with the bottom probe indicating anoxic conditions (oxygen levels below the sensor detection limit of <0.012 mg

L⁻¹), and ΔS decreased, indicating reduced vertical mixing. This demonstrates that initial fluid mud formation can induce local

stratification, even during overall weakly stratified conditions (Becker et al., 2018; Becker et al., 2013).

During the persistent state (Fig. 7c), fluid mud occurrence was continuous, with cumulative duration across both ebb and flood

tides of approximately 8 h and TSS values > 10 g L⁻¹. TSS increased rapidly from 0 to > 30 g L⁻¹ within 5 – 10 min after

current velocities reached their minimum. Since the mid probe recorded TSS > 10 g L⁻¹, the fluid mud height was estimated

to exceed 2 m. The dissolution of the fluid mud layer occurred more gradually than its formation, following peak current

velocities (> 100 cm s⁻¹) within an hour after LWS. Strong vertical oxygen gradients persisted, with sharp drops at HWS, while

ΔS increased due to intrusion of saline marine water. This indicates that fluid mud layers not only influence local oxygen



depletion but also modulate salinity stratification during flood tides, as dissipation is dampened (Oberrecht, 2021; Wu et al., 2023; Becker et al., 2018).

During the break-up phase (second ebb tide , time[h] = 12–18 h), TSS values gradually decreased below the 10 g L⁻¹ threshold.

Oxygen concentrations remained low but did not indicate full anoxic conditions, and ΔS values indicated onset of vertical decoupling, peaking around LWS. The synchronicity of salinity stratification with LWS suggests that tidal timing critically affects fluid mud stability and vertical mixing (Winterwerp et al., 2017). By the third HWS (time [h] = 24), oxygen values recovered and were consistent throughout the water column.

Over multiple spring-neap tidal cycles, four recurring fluid mud states were identified: build-up, persistent, break-up, and

absence. Build-up and persistent states coincided with spring tides, while break-up and absence were more associated with neap tides This contrasts with most previous studies, which report enhanced bed shear stress, entrainment rates, and SPM concentrations during spring tides, yet do not fully resolve the short-lived fluid mud dynamics observed here (Dijkstra et al., 2024; Manning et al., 2010; Wu et al., 2023; Dijkstra et al., 2025).

We also observed a key discrepancy: the temporary mooring did not detect fluid mud during neap tide, despite its presence in

the semidiurnal cycle in our measurement campaigns. This likely reflects differences in vertical resolution and lateral positioning, highlighting the highly dynamic and spatially variable nature of the estuary, which can lead to divergent interpretations. Mooring data showed rapid TSS increases and oxygen drops around HWS, suggesting sensors were rapidly covered by an anoxic, hyper-turbid layer within a single 5-minute interval. This implies that fluid mud formation and resuspension can occur on timescales shorter than typical monitoring intervals, emphasizing the importance of high temporal

resolution observations (Abril et al., 2004).

Stronger tidal forces during spring tides, with increased tidal amplitude, bring a larger volume of marine water into the estuary, which is typically richer in oxygen compared to the resident water (Webb and D'Elia, 1980). This likely promotes the availability of degradable organic matter, which can support microbial activity and nutrient cycling (Defontaine et al., 2019; Manning and Dyer, 2002; Winterwerp et al., 2017). Organic matter can act as binding agent between mineral particles (Zander

et al., 2022b), promoting the formation of flocs that enhance fluid mud accumulation and prolong the persistence of fluid mud layers, as observed in our study during the persistent state linked to spring tides (Fig. 5).

Following sedimentation, the degradation of sedimentary organic matter (SOM) over time may reduce fluid mud stability (Talke et al., 2009a; Zander et al., 2022b; Mudge et al., 2007). Previous studies attribute a decrease in yield stress, the critical stress to initiate fluid-like behaviour, to microbial decomposition of SOM (Zander et al., 2023; Zander et al., 2022a; Zander et

al., 2022b; Gebert and Zander, 2024). When SOM is degraded or the cohesive organic components break down, sediment consolidation may increase density, leading to a more solid structure. In our observations, although initial fluid mud density was higher during neap tides, the layer exhibited lower stability and increased susceptibility for vertical mixing compared to spring tides (Fig. 7, Tab. 2). This is consistent with the notion that lower yield stress reduces interparticle friction, making the sediment easier to resuspend (Zander et al., 2022b; Fernando, 1991; Tu et al., 2019).



During neap tides, the less stable fluid mud layer appears more susceptible to entrainment, which is supported by increased turbulence, shear stress, and current velocity in the water column above the lutocline (Fig. 5, Fig. 7). This counterintuitive increase in flow intensity has been attributed to the reduction of the effective hydrodynamic cross-section by the relatively immobile fluid mud layer, which constrains flow and accelerates currents (Talke and Jay, 2020). Consequently, portions of the fluid mud layer may be resuspended or redistributed laterally and longitudinally toward the fairway or upstream after neap

tides, depending on river discharge and flow conditions (Li et al., 2023; McSweeney et al., 2016; De Jonge and Schückel, 2019).

Overall our observations suggest that the persistence and stability of fluid mud are closely linked to tidal amplitude, SOM content, and hydrodynamic forcing, with spring tides favouring longer-lasting more stable fluid mud layers, while neap tides promote increased mixing and potential resuspension.

**5 Conclusion**

Understanding hydro- and sediment dynamics in hyper-turbid estuaries is crucial due to their ecological and socio-economic importance. In the Ems Estuary, fluid mud exhibits strong vertical and temporal variability, affecting navigation, oxygen dynamics, and overall ecosystem function.

Our observations reveal two superimposed temporal cycles: short-term variability over the semidiurnal tidal cycle, and longer-

term modulation by spring-neap tides. Spring tides favour persistent, stable fluid mud layers, while neap tides lead to reduced stability, enhanced turbulence, and resuspension or lateral redistribution of the mud layer. These patterns are primarily controlled by tidal amplitude, current velocity, and the hydrodynamic reduction of cross-sectional area caused by dense fluid mud layers. Biogeochemical processes, such as SOM decay, may also influence layer stability, but their role requires further investigation.

This study highlights the importance of long-term, high-resolution observations to capture the full dynamics of fluid mud. The results demonstrate the critical role of spring-neap cycles in regulating hydrodynamics, turbulence, and fluid mud stability, and emphasize the need for more interdisciplinary approaches that couple physical drivers and biogeochemical processes to improve modelling and management of hyper-turbid estuarine systems.



**Data availability**

Data of temporary mooring and measurement campaigns will be made available via Zenodo (doi: 10.5281/zenodo.17634879) upon publication of the manuscript. Data will be linked to the geoportal.bafg.de of the Federal Institute of Hydrology.

**Author contributions**

**Conceptualization:** AS, CB, LR, TH; **Data curation:** AS; **Formal analysis:** AS, CB; **Funding acquisition:** CB, AF, OR; **Investigation:** AS, CB, AF, LR, JL, DH; **Methodology:** AS, CB, TH; **Project Administration:** CB, DH, AS; **Resources:** CB, TH, AF; **Software:** AS; **Supervision:** TH; **Validation:** AS, CB, TH; **Visualization:** AS; **Writing (original draft preparation):** AS; **Writing (review and editing):** LR, DH, JL, AF, OR, CB, TH.

**Competing interests**

The corresponding author declares that the authors have no competing interests.

**Acknowledgements**

We gratefully acknowledge the Waterways and Shipping Administration Ems-Nordsee for their support in facilitating the measurements, maintaining the equipment, and providing access to the data. We especially thank the crew of the R/V *Friesland* for their dedication and invaluable support throughout the measurement campaigns, which made the data collection both possible and rewarding. We also thank the technical and laboratory staff at the Federal Institute of Hydrology for their committed contributions to the data collection.

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
