# Peer review of "Vertical and temporal fluid mud dynamics during spring-neap tidal cycles"

_EGUsphere, 2025_

## Referee Comment (RC1)

The authors present an analysis of two sets of observations of, among other things, sediment concentration and velocity in the hyperturbid Ems river. Their aim is to show the fluid mud (FM) dynamics on tidal and forthnightly timescales, where especially the forthnightly aspect is new, as well as the different tidal dynamics during spring and neap. On a tidal scale, they identify various stages and describe the ebb-flood differences during spring and neap. On the forthnightly scale they identify FM building up and breaking down with the spring-neap cycle.

I think the subject of this manuscript is really interesting. Describing fluid mud dynamics on various timescales is important for the field, especially since models are not yet able to reproduce this dynamics. I really like that the authors bring in that the reduction of the effective cross-section by fluid mud is an important aspect. I also think it is valuable to have such detailed data describing multiple tidal cycles, thus complementing the study by Becker et al (2018) greatly for local knowledge on the Ems.

I do think however the authors could be making a mistake in their analysis. They are implying the sediment concentrations they observe are primarily concerned with vertical dynamics. This could be reasonable if they were measuring in the centre of the ETM. However, when I checked the NLWKN stations for the time of the 2019 mooring, I found that this was around a discharge of 20-25 m3/s and the ETM was strictly upstream from Leerort. This means the measurements were taken right at the edge of the ETM and movement of the ETM could be important both during the tidal and the spring-neap cycles. Also in 2023 it seems that the ETM was quite far upstream but less so than in 2019. This could explain the neap absence of FM in 2019 and the fact that fluid mud was still there in the 2023 neaps. I'm wondering if this also could explain why Becker et al (2018) did not see FM disappearing during a few hours after LWS while the authors did. Given the above, I think the authors need to consider if along-channel dynamics could be important.

Also, I think the authors are not very clear about the exact research aim, especially on how this complements existing literature and what the knowledge gap is. They do mention the relevant literature though, so this is mainly about better formulating the novelty of their study. Furthermore, I found the results in section 4.1 quite confusing and seemingly contradictory with each other and with Becker et al (2018), and I think this section needs clearing up and a good discussion summarizing if all results are consistent or not. I have some further comments/questions related to the equations and analysis methods from section 3.1 as well. All my comments are described in detail below.

I think this manuscript could be valuable to the research community but given the possibly big gaps in the reasoning, I would say that very major changes are needed. Kind regards, Yoeri Dijkstra

**Main comments**

1. Introduction

After a general introduction to the relevance of mud, the authors on ln 44 go into terminology, characteristics and definitions of FM. Temporal variability – the main topic of this work – only appears on ln 68. It would be easier to read if the authors started from this and introduced the necessary terms from there. Some of the terminology and definitions may be omitted. While the authors discuss the many different properties and definitions used by others, their definition (and motivation for this) remains somewhat implicit (I guess line 63 gets closest). Please state your definition.

The Ems is suddenly mentioned on ln 69 and again ln 83. It remains unclear though whether the Ems is topic of this study and whether the study should be considered general or specific. Please introduce clearly that this study is about FM in the Ems, with the idea that dynamics carry over at least in part to other highly turbid tide-dominated estuaries. Please also state that results are for almost fresh water and strongly dominant tides.

While the aim of the study is stated on ln 98-99, I found the knowledge gap unclear. Clearly, Becker studied tidal dynamics in great detail in the same river and ln 77-82 describe spring-neap dynamics, so what is missing? I think it is really important to make this explicit, as this study now seems to promise more of the same, which understates the results.

2. 1D model and equations

I did not understand the 1D model and Equations 1-6. Firstly, the symbol z seems to be used for total depth, while it was defined (ln169) as the vertical profile. Since all depths are scaled, total depth would be unity. E.g. the '1d model' on ln 173 is $v=q/(fz)$ should be $q/(f*depth)$, right? Here depth should also be with dimension m to make sense. Also z appearing on ln 171 and eq 1, 3 and 5 did not make sense to me; supposedly Ri is a function of h, but 'z' appears on the right-hand side. Please ensure notation is properly introduced and dimensions used or omitted consistently.

Secondly, Fig 3 should be based on a 1D model, i.e. ln 173, but it took me quite some effort to see how. The 'model' seems to say the same amount of water needs to go through a smaller cross-section if the FM thickness is bigger. To improve clarity, please give the model its own equation number, and near Fig 3 please rework the equation into an explicit expression for h_L in terms of v and state that q was fitted in the main text. Given the simplicity of the model, I suggest not to interpret differences between spring and neap in Fig 3 – given uncertainty, the spring and neap curves could well be the same.

Given the complex dynamics described in the rest of the manuscript, '1d model' has the risk of being mis-interpreted as some numerical fluid dynamics model. Terming it '1D mass balance' or otherwise emphasizing that the model is only a mass conservation principle also near Fig 3 would be helpful.

Eq. 1-4 it seems define various quantities from any level (h or z) to the 10g/L level. So close to this level, it is almost based on the derivatives d/dz, while far from it, they are almost bulk quantities. Both have different interpretations, so I don't see how this works. Also, in Fig 4 I see RI, N2 and ue computed at high vertical resolution, so this seems to be done differently from these equations. I'd much rather see formulations based on derivatives or a small interval dz of the smoothed vertical observations, as this is how the data is interpreted in section 4.1.

I struggle with Eq. 3. The ue is defined here based on u* and a bulk Ri, which are both quantities that should (by definition) not be functions of a vertical position. However, ue is a interpreted as a function of the vertical position. I looked up that Kranenburg and Winterwerp defined this as bulk quantity as well that should not be interpreted over the vertical.

Also in Eq. 4, u* should not depend on z. Usually it is determined close to the bed assuming a log-profile exists there. Here a modification could be u*=kappa*1/2*(hbed+h_L) * (uh-ul)/(h_L-hbed), where hbed is the bed level. Still, this would project the log layer over the water column up to h_L which is quite a stretch.

3. Section 4.1

I got confused by the text in this section. Some of the claims about higher Ri and/or N2 I could not see from Fig 3 and lines 270-278 seemed to partly contradict ln 279-287, although this may be because the authors mean to discuss different parts of the tidal cycle. I will list my comments per line below, but would encourage the authors to rewrite this section, starting each paragraph with exactly what stages, subfigures and tables they talk about. Also from ln 288 the results are reiterated in a discussion style and it would be easier to read if the authors also explicitly stated the topic/purpose of the paragraph in the first line.

- Ln 238: 'around LWS'. This is 2h after LWS right?
- Ln 239: 'ebb' -> late ebb
- Fig 2: why do you not find (near-)zero velocity at LWS?
- Ln270-271: for spring, yes. For neap, the entrainment in stage 4, 5 is similar to stage 1

- Ln272: 'as the FM layer thinned': this is a matter of timescales; from stage 2,3, the FM layer thickens again in stage 4,5. Do you mean thinner than during ebb here? I found this ambiguous.
- Ln276: 'RI became very low': what stage are you thinking about here? Fig 4, stage 5 shows lowest velocities, but RI nor entrainment are very low, so I really cannot see where this claim comes from.
- Ln 282-287: you are talking about ebb here but referring to fig 4b, which is stage 2 and hence early flood. I got confused here.
- Ln 282-283: I don't see the >50g/L being thicker during neap tide (fig 4a,4b). The spring and neap lines indicating 50 g/L are at a similar relative height. Maybe in terms of absolute height this means the neap one is thicker near LWS, but then indicate this and why absolute height now is discussed instead of relative.
- Ln 285: 'limited vertical exchange'. This is about ebb neap tides. In ln 270 it was claimed that entrainment was relatively high during ebb, so this feels inconsistent. I don't understand this anymore at this point.
- Ln 286: the FM layer mentioned here now points to stage 1,2 and the 10-50 g/L layer? Or is this still about the >50g/L layer. I'm not sure anymore.
- Ln 286: also, on previous lines the *thickness* of the layers was discussed. Now it is suddenly about density. I found it hard to follow the switch and understand the message of this paragraph.
- Ln 286-287: Maybe Ri is locally lower for spring stage 1,2 compared to neap, but the velocity profile seems more sheared and the N2 similar or bigger indicating less mixing. I'm not so sure how to read this and how to assess the spring neap differences from this section.
- Ln 288: One sentence earlier it was said that spring had more turbulence in stage 1,2. Apparently the focus has switched again.

The authors identify 5 stages. I found it unfortunate that these are different from the 5 stages by Becker et al. Becker's stages 4, 5 seem to be stages 1, 2 here. Then Becker's stages 1,2,3 all had a similar FM layer thickness, but different vertical structure (mixed, restratified, stratified with no flow in lower layer, respectively). This structure is not studied in the present manuscript so a differentiation in stages 4,5 here was not apparent to me; these stages look the same in Fig 2. Remarkably, the present manuscript finds FM disappears briefly (stage 3). This was not observed by Becker et al, but the authors do not comment on this. The authors are free to define their own stages, but a comparison with Becker et al would be in place.

The authors assume the dynamics is primarily vertical. In the comment below on section 4.2 I suggest that along-channel or lateral dynamics cannot be ruled out as important. I'm

fine with the authors not discussing this much in this section, but I'd like a bit more emphasis on the assumption of vertical dynamics and a small remark in the discussion stating that lateral/along-channel dynamics could play a role.

4.  Section 4.2 and the importance of along-channel dynamics.

The first part of the section describes an interesting asymmetry in the spring-neap cycle, with FM not present during part of the neap tides. This is different from the campaigns discussed in section 4.1. The authors seems to assume that the dynamics observed here should be consistent with that of the campaigns 4 years later. They don't seem too confident explaining the difference though: ln 351-352 relates it to minor variations due to local conditions, while ln 434-436 also mentions local conditions, as well as vertical resolution. I struggle with these explanations, since section 4.1 showed FM coming up quite high in the water column every tide. Sure, on a different location this could be a bit more or less and FM could stay under the sensor, but consistent absence even 1 m above the bed in a similar close-by cross-section looks unlikely.

I would suggest another potential contestant: along channel dynamics. I looked at the observations from the permanent measurement stations and a simple reconstruction based on linear interpolation in the along-channel direction gives the *maximum* concentration during the tide on Aug 27 2019 (neap tide) in Fig 1 below. In Leer, the maximum reconstructed SSC in the thalweg (based on a very simple vertical model) is ~ 15 g/L, so sensibly 10 g/L is not reached a bit on the flanks. The FM is not gone though, but just slightly more upstream, and measuring in Weener could have provided a different result. It could still be that vertical dynamics is important here, but it also seems that you are measuring right on the edge of the ETM, so along channel movement of the ETM could also be the key here.

Similarly, the along-channel dynamics could maybe explain that FM is 'only' present up to 60-80% of the time during springs. Is the FM consistently gone during a later part of the flood tide and only coming back on the ebb? That could indicate upstream advection.

In my opinion, Fig 5b and 6b do not contain new information and I found it quite hard to read. I'd prefer fig 5a and 6a in one figure plotted underneath each other for better overview (Just a suggestion and up to the authors to choose what to do of course). Also the use of colour in Fig 5a and 6a was confusing to me since it does not carry new information; it is simply demarcating when the bars drop below 0.5. The 0.5 is an arbitrary criterion, so I'd find it more objective and clear if it were one colour.

[Figure]

*Figure 1: reconstruction of an along channel section along the thalweg and based on the NLWKN stations on 27 Aug. 2019, using the maximum measured SSC at the stations during over the tidal cycle. Own preliminary results by the reviewer.*

From ln 388, or actually from ln 383, I suggest to start a new subsection. I was quite confused about the switch from spring-neap dynamics to again describing tidal phases. Please introduce that you will switch focus and why; you already discussed tidal dynamics in section 4.1 so what point do you want to make coming back to it.

It struck me how small the ebb velocities were and cannot see how ebb and flood water transports balance locally. This indicates that the lateral dimension is quite important for the flow. I think it is worth mentioning this with some emphasis.

5. Organic matter

Organic matter makes an entrance in the manuscript on ln 443-454. No observations of this are shown. Yet, ln 462-463 states that the 'observations suggest' that FM is linked to organic matter content. Also the organic matter features prominently in the conclusion. I think both sentences should be removed. If the authors wish to hypothesise about organic matter in relation to literature, I suggest to do this in a separate discussion section, not together with the results, and it should not feature in the conclusions.

6. Conclusions and abstract

The conclusions and abstract now almost completely skip the tidal results, which is about half the paper. The conclusions then say about neap tide that the water column is less stable, while the abstract states that the majority of the water column is filled with FM, but neither discuss the absence of FM during neap in the mooring. This to me feels like cherry-picking results, while not mentioning other results that show the story is much more nuanced and complex. I would like to see the conclusion/abstract fairly reflecting the complexity of the results.

**Minor comments**

- I found some references missing in the bibliography, e.g. Dijkstra et al 2018, Nakagawa et al 2012. Please check all references.
- Ln 30 'up to several g/L' -> well over several g/L(?)
- Ln 33: *high* SSC does not indicate *raising* costs. *Increasing* SSC might.
- Ln 64: turbulence damping would already occur at 0.1-1 g/l, so this feels a bit misleading
- Ln 65: non-newtonian behaviour only matters >> 10g/L, so in your apparent definition FM is not necessarily non-newtonian.
- Ln 86-88: the link with organic matter was not really made for the Ems yet and its importance is, I think, still subject of discussion (see e.g. Horemans et al 2021, doi: 10.1029/2020JC016805 suggesting it might not be too important for the Scheldt). All references here seem about the Elbe. Since you are talking about the Ems, this claim seems out of place.
- Table 1: this table is big and, though interesting, feels very much out of place. Your manuscript is not about various definitions of FM and actually the exact definition does not matter too much for your work. I'd suggest removing it (or moving it to an appendix) and less stressing FM definitions in the text. I'd rather see your exact definition and motivation, since that is currently missing.
- Ln 83-84: be sure to write SPM *concentration* here. SPM mass is not at its maximum for lowest discharges.
- Ln 85: you conclude here that FM is absent in winter. Your reasoning just states that SPM concentrations are low during this season, while Winterwerp 2017 suggests that FM may persist – whether you believe this or not, I suggest avoiding this discussion here.
- Ln 120: Pein et al 2014 do not talk about SSC. De Jonge et al 2014 is a better reference here.
- ln 128-129: For this statement it is important that you also explicitly mention your upper bound on FM concentration. From personal communication with Dennis Oberrecht, I figure he observes stationary layers of mud of around 200 g/L and 500 g/L with some different behaviours. If I understood correctly, especially the 500 g/L first grows under increasing discharge (before eroding at very high discharge), so I guess any upper bound under 500 g/L will work.
- Ln 149: please state exact dates of the campaigns.
- Fig 2: I'd expect HWS 4-5 after LWS looking at Becker et al. This would be consistent with your velocities. Could you indicate HWS as well?

- ln 260: here and elsewhere you talk about 'hydrodynamic control': what does this mean precisely?
- Ln 281: 'generally considered quasi-stationary': this term does not appear in table 1. I'd argue that the 50g/L limit is a bit flexible. Do you mean above gelling? Mentioning you expect this layer to be around/above gelling point, you could claim it is quasi-stationary without reference I think.
- Section 4.2: please mention here you are switching from the campaigns to the mooring.